# Encapsulation of Essential Oils for the Development of Biosourced Pesticides with Controlled Release: A Review

**DOI:** 10.3390/molecules24142539

**Published:** 2019-07-11

**Authors:** Chloë Maes, Sandrine Bouquillon, Marie-Laure Fauconnier

**Affiliations:** 1Institut de Chimie Moléculaire de Reims, UMR CNRS 7312, Université Reims-Champagne-Ardenne, UFR Sciences, BP 1039 boîte 44, 51687 Reims CEDEX 2, France; 2Laboratoire de Chimie des Molécules Naturelles. Gembloux Agro-Bio Tech, Université de Liège, 2 Passage des Déportés, 5030 Gembloux, Belgium; 3SFR Condorcet FR CNRS 3417, Université Reims Champagne Ardenne, UFR Sciences Exactes et Naturelles, BP 1039, 51687 Reims CEDEX 02, France

**Keywords:** essential oil, encapsulation, controlled release, biopesticide

## Abstract

Essential oil (EO) encapsulation can be carried out via a multitude of techniques, depending on applications. Because of EOs’ biological activities, the development of biosourced pesticides with EO encapsulation is of great interest. A lot of methods have been developed; they are presented in this review, together with the properties of the final products. Encapsulation conserves and protects EOs from outside aggression, but also allows for controlled release, which is useful for applications in agronomy. The focus is on the matrices that are of interest for the controlled release of their content, namely: alginate, chitosan, and cyclodextrin. Those three matrices are used with several methods in order to create EO encapsulation with different structures, capacities, and release profiles.

## 1. Introduction

### 1.1. Essential Oils

Essential oils (EOs) are volatile liquids extracted from various organs of plants. Although plants have been largely used in medicine, agriculture, and perfumes for many years, interest in their EOs has massively increased in recent years, thanks to their incredible properties. Indeed, numerous studies have shown antioxidant and biocide activities, which are usable in many areas [1,2]. EOs are biosynthesised in aromatic plants as secondary metabolites, and are mainly composed of terpenes, but also contain other chemical compounds. Most of the time, the chemical composition of EOs is very complex and consists of a mix of isoprenoids (monoterpenes and sesquiterpenes) that could be represented by one major compound or different compounds in equal proportion [3].

EO extraction methods are specific to their hydrophobic and volatile nature. Conventional methods include hydrodistillation and steam distillation for most plant parts, and cold expression for citrus peel [4]. Some other innovative techniques exist for obtaining the hydrophobic and volatile fraction of plants, such as supercritical carbon dioxide fluid (scCO_2_) extraction or microwave-assisted extraction [5]. Even if the same fraction is obtained, these products cannot be called EOs, because standard (ISO 9235) reserves this term for products obtained by conventional methods [1,6,7]. The generic name given to these hydrophobic and volatile products is “plant extract” (preceded by the name of the technique used, so as to be more specific) [8]. Technological progress has improved the efficiency of these innovative techniques over time, but steam distillation still remains the most used process, because it is efficient, green, and inexpensive [3,9].

As previously mentioned, the main component of the chemical composition of EOs is a complex mixture of hydrocarbon terpenes and terpenoids. The majority of the first group consists of monoterpenes and sesquiterpenes, and the second group consists of oxygenated derivatives of hydrocarbon terpenes [1]. It has been shown that several criteria impact this composition. First, the EO of one species can vary, depending on chemotype. A chemotype is a “chemical race”, meaning that plants sharing the same botanical name (same genus and species) can exhibit a completely different chemical composition. The EOs’ composition can also vary depending on the organ extracted [10]. Secondly, the same plants growing in different places can produce EOs with a slightly different chemical composition. Indeed, the soil composition, exposition, climate, cropping practices (pre- and post-harvest), rainfall, and presence of insects or other biotic and abiotic stress all influence the composition of the EO [3,11,12,13,14].

The interest in EOs is explained by their large number of biological activities [15]. A lot of research has been carried out in order to determine their antimicrobial, insecticidal, antioxidant, and herbicidal effects [16,17,18,19,20,21]. For example, antimicrobial activity against *Escherichia Coli*, *Staphylococcus aureus*, and *Bacilus subtilis* has been found for *Citrus medica* L. var. *sarcodactylis*, cinnamon, thyme, and citronella EOs [22,23,24,25]. Antifungal properties have also been reported; *Penicillum parasiticus* and *Aspergilus niger* growth has been inhibited by *Tetraclinis articulata* (Vahl) [26]. Strong antioxidant activities have also been found, for example, in *Juniperus phoenicea* and *Citrus medica* L. var. *sarcodactylis*, EOs [22,27]. Some specific insecticidal effects have also been displayed by *Juniperus phoenicea* EO against *Tribolium confusum* in stored seeds and *Cymbopogon citratus* (DC.) Stapf, *Ocimum canum* Sims, *Ocimum gratissimum* L. var “gratissimum” L., and *Thymus vulgaris* L. EOs from Cameroun against anopheles gambiae Giles [27,28]. Another interesting effect was the allelopathic inhibition of *Amaranthus viridis* seed growth by *Eucalyptus terticornis* EO [29]. All of these activities could be utilized to create biopesticides [30].

A lot of other applications exist for EOs, such as their use for improving human health and wellbeing [31,32]. EOs are widely used in aromatherapy, cosmetics, and massage [33]. From an industrial point of view, EOs are also used as preservatives and flavors in food [34], as well as in fragrances in soaps and perfumes since the 19th century [3,35].

### 1.2. Controlled Release

The control of high volatility of EOs is the main challenge that has been solved by the development of various encapsulation techniques. Encapsulation can retain EOs by a physical or chemical interaction with a matrix, which holds the essential oil for a longer time [3]. Almost all applications of EOs require an increase in retention times and different release profiles [1]. For example, in cosmetics, it is appropriate to develop a method of encapsulation in which the EOs are released by mechanical effects [36]. On the other hand, the encapsulation of flavors for food applications calls for a moderate controlled release [37]. Depending on the final application, it will be necessary to refer to the legislation in force in the field for the choice of matrix to encapsulate EOs (e.g., allergens in the cosmetic field), or to carry out additional toxicity/allergenicity tests [38].

In the case of pesticides, encapsulation needs to allow for a slow and continuous release of the active agent at an optimal threshold in the environment [39]. The minimum release rate is determined by the efficiency of this active agent, and the maximum by its phytotoxicity scale. Figure 1 shows the ideal outline of a biopesticide controlled release over seven days, which consists of an initial rapid release at the middle concentration between the efficiency and toxicity scale, followed by a long and constant release. A lot of parameters are involved, because of the level absorption by the plant, volatilization, leaching, and degradation [40]. Another important point is that applied pesticides need to be available at the optimal rate for at least one week in order to avoid necessitating repeated applications [41]. The release of the active agent after encapsulation is possible by simple diffusion, exterior factor intervention, or matrix erosion. In each case, it is important that neither the matrix, nor its degradation residues, are toxic to the environment. As a consequence, many natural polymers have been used to encapsulate pesticides.

The special case of EOs’ controlled release for biopesticides will be detailed in this paper, after a short presentation of the different encapsulation techniques that allow for EO retention.

## 2. Essentials Oils Encapsulation Techniques

As previously said, it is important to protect and retain EOs in order to extend their shelf life and activities. For this purpose, a large number of encapsulation methods have been developed in recent years. These methods allow for slow release and are summarized in Table 1. Seven major techniques have been the most studied for the creation of EO particles or capsules. Indeed, for the encapsulation, four types can be listed, as follows: (i) particles generated by a matrix where EOs are dispersed; (ii) capsules with a membrane surrounding a core where the EOs reside; (iii) complexes, where EOs are stabilized in cavities by chemical interactions; and (iv) droplets created by a simple emulsion in surfactants (Figure 2). Each encapsulation type can also be characterized by the size of the generated objects, namely: “micro” (1–1000 μm) or “nano” (<1000 nm) [42].

In Table 1, the name of the techniques employed (emulsification, coacervation, spray drying, complexation, ionic gelation, nanoprecipitation, and film hydration method) are presented in the first column; in the second column, we present the nature of the generated objects (droplets, particles, capsules, and complexes). The third and fourth columns briefly detail the methods and the average size (micro- or nano-scale). Finally, some examples of the corresponding matrices are presented with their references. At this stage, it is important to note that some matrices (alginate, chitosan, and cyclodextrin) can be used for many techniques.

## 3. Particular Candidates to Encapsulate EOs in Order to Facilitate a Controlled Release

The choice of the EO encapsulation technique and matrix is made depending on the required application. In the case of the creation of a bioproduct, the release profile has to be studied with different methods.

The first method to study the release is the analysis of the targeted bioactivity, which directly shows the methods of encapsulation in their application context, but does not measure the quantitative aspect of the release [76]. These analyses can be realized through two different ways—in open (in situ) or closed space (in vitro). The in-situ way shows the real interest of the EOs encapsulation, as this represents the actual conditions. For example, the efficiency of the EOs encapsulated for their antimicrobial activity can be determined by studying the effect on the growth of bacteria. These assays can be performed in petri dishes, which are a closed space called “in vitro”, or can be on a targeted product (e.g., on a cake in the development of natural preservative in bakery), which are the real conditions called “in situ” [47].

A second and strictly quantitative method to study the release is the use of gas chromatography coupled with mass spectrometry, in order to measure the quantity of EOs that can be released during a specific period of time. This method can be used with a static headspace (analysis of a sample of headspace) [57] or dynamic headspace (continued intake of gas flow, trapping on adsorbent cartridges, thermal desorption, and cryofocusing) [92].

Each method is of interest, giving different pertinent information, as follows: a biological activity or a quantity of EO released per unit of time [57]. In addition, the release profiles are influenced by several factors such as the pH, temperature, humidity, and component concentrations; therefore, it is quite difficult to compare two studies about encapsulated EOs’ release, and a nuanced observation should be done.

In Table 1, three matrices have been used several times to encapsulate EOs. Alginate, chitosan, and cyclodextrin seem to be particularly interesting thanks to their ability to be used with different encapsulation methods, and thanks to their natural origin. The release studies will be presented independently.

### 3.1. Alginate

Alginate is found five times in Table 1. The first associated method is multiple emulsion [45], but this was then combined with the most important one—ionic gelation [74,78]—because alginate has the ability to create an “egg-box” structure in the presence of calcium chloride. Ionic gelation with a simple emulsion allows for a controlled release, but this was influenced by the calcium chloride concentration and the cross-linking time—the increase of both factors reduced the steady state of EO release, but after 150 hours, all of the EOs were released [74]. The release from encapsulation by the combination of multiple emulsion and ionic gelation was also studied by in vitro analysis, showing an initial burst effect of nine hours, followed by a second slower stage for 16 hours [78]. Recently, Riquelme [83] developed active films that were constructed with a matrix based on the ionic gelation method, and then covered with a plasticizer in order to preserve food against microorganisms. The last method that could be employed is spray drying—the combination of alginate and cashew gum showed a longer performance, where 45% of the EOs were released within 30 hours and 95% in 50 hours [52].

### 3.2. Chitosan

As summarized in Table 1, chitosan is used to achieve encapsulation with four different methods, giving four different products. First, the method of Chen [88] gives a nanogel with very strong covalent linkages. As a consequence, the release of EOs is quite slow. This method used by Zhaveh [86] showed that one week was necessary in order to release 78% of *Cuminum cuminum* EOs in a chitosan–caffeic acid nanogel, and one month was necessary for a total release. Similar results were obtained by Beyki [87] with *Mentha piperita* EOs in a chitosan–cinnamic acid nanogel. Also, both had an enhanced antimicrobial activity against *Aspergillus flavus* in sealed and non-sealed conditions.

The second well-used method is the combination of emulsion and ionic gelation to generate nanoparticles [78]. The use of sodium tripolyphosphate allows for ionic crosslinking with divalent cations. These nanoparticles are very regular, separated, and stable, but the release profile is different when compared with the first case. Indeed, for all of these types of nanoparticles, an initial burst effect followed by a slow sustained release was observed [75,76,93]. It was also found that the releases were faster in acidic conditions than in neutral ones [75,76]. Even if the total time of the release is also approximately one month, the differences in the release profiles could influence the applications. Mohammadi [76] explained the interest in encapsulated EOs as an alternative to synthetic pesticides at the pre- and post-harvest stages in fighting against the fungus *Botrystis cinereal* (the causal agent of grey mould disease). Ghaderi-Ghahfarokhi [23] used the same techniques to create a nano-formulation of liquid additive for food, which improved the solubility and stability of cinnamon EO, in addition to controlling the odor and release of the compound. The release profile was studied in real conditions (beef patties) that gave a faster release (total after 104 hours), but still in two stages.

The third encapsulation technique using chitosan consists of generating nanoparticles by nanoprecipitation. These nanoparticles are smaller than those prepared by ionic gelation, but showed a good antibacterial activity, especially against *Staphylococcus aureus*. This technique also allows for a slow release, but the inhibitory effect against bacteria is lower [24].

Finally, chitosan has also been used with the spray drying method. Dima [53] showed that the pH has an influence on the release rate, and, for example, chitosan nano-capsules had less retention at a of pH 2.5, while alginate nano-capsules had the same effect at a pH of 6.5. The best performances showed retention for 10 days.

In conclusion, chitosan is a good natural product to encapsulate EOs, because it can be used in many techniques, although the release profile appears difficult to control and/or modify. Some further optimizations could still improve these performances.

### 3.3. Cyclodextrins (CDs)

For cyclodextrins, three technics have been found. In 2003, the “kneading” method was developed by Manolikar and Sawant—EOs and CD were mixed in a mortar and kneading with a small quantity of ethanol, and the resulting paste was dried under vacuum at −20 °C so as to generate a fine powder [61]. This method was also used by Santos, who showed that kneading complexes of carvacrol and β-Cyclodextrin (β-CD) had a high entrapment efficiency, enhanced antimicrobial activity (*S. typhimurium* and *E. coli* K12), and good stability under lighted storage conditions [62,63].

Dry particles of β-CD encapsulations can be obtained by the freeze-drying (FD) method of Karathanos V.T. et al., which consists of the freezing and lyophilisation of the EO-(β-CD) complex [63,67,68,69,70]. The encapsulation of the compounds induces the displacement of water from the internal cavity of β-CD, influencing their stability relating to the relative humidity of storage [71]. For carvacrol encapsulation, the FD method led to a higher encapsulation efficiency than the kneading method [62]. Recently, Da Rocha Neto et al. studied the influence of each encapsulation parameter of the freeze-drying technique, and determined that the encapsulation efficiency can vary from 1% to 70%, depending on the characteristics of the β-CD (intramolecular water content) and the EOs (type and concentration), but also on the method parameters (adding order of components, drying process, and solvent used for the extraction). The factors influencing the release kinetics of the EOs were also studied, so that the temperature and the relative humidity (RH) showed a significant effect as attempted. Indeed, the lowest release was obtained at 4 °C and 0% RH, although the highest release of EOs was at 40 °C and 45% RH for palmarosa EO, and at 23 °C and 98% RH for star anise EO [72]. Kfoury et al. also showed that the EOs released from CD inclusion can be controlled by temperature, RH, and preparation method. In general, the profile release of eugenol is almost constant for its entire duration, and the half-time of release was 228 hours at 60 °C, whereas it was 69 hours at 100 °C [65]. To go further in the understanding of EOs’ inclusion into cyclodextrin, some nuclear magnetic resonance spectrometry analysis showed interactions corresponding to two possible inclusion modes. In addition, they showed a strong antifungal activity in unsealed conditions [66,73].

Another technique consists of simply blending EOs and cyclodextrin in water [57,64]. Despite the fact that the retention of EOs by cyclodextrin, in this case, is not a real encapsulation, but more a complexation, the results of Kfoury presented an interesting release profile, probably because of the suitable cavity size of the β-Cyclodextrin that had a better retention capacity for EOs. The release rate is an exponential asymptotic trend and 10% of EOs are released after 135 min [57].

## 4. Discussion

The goal of this paper was to assess all of the techniques and matrices developed until today, for encapsulating EOs, in order to achieve a controlled release. The first part presented all of the lab techniques usable, with various matrices, and the second part detailed more information about the most used matrices for controlled release. As previously mentioned, their application as biopesticides is very complex, because many cases and settings are possible. A continuous release at the optimal threshold for at least one week is essential, but the most important requirement is to preserve this release in natural conditions. Parameters such as the absorption by the plant, volatilization, leaching, and degradation have to be studied in detail for each product. Considering the great diversity of potential applications, it seems that each method and each matrix may have a specific use.

First, focusing on the encapsulation, we can observe that each technique can be used for a number of matrices, and also the opposite, that each matrix can be used with a number of techniques. The combination of both a specific encapsulation method and matrix creates a well-marked product, such as a particle or capsule, micro- or nano-particles, physical or chemical interaction, and so on. In addition, as EOs are highly variable, many different encapsulation products can be found. Lots of articles study only one or two of these products, with a lot of information about the encapsulation efficiency, kinetics of release, and biological properties.

On the technical side, the major difference between the encapsulation methods is the products’ physical aspect—some give a dry powder (spray drying and nanoprecipation), others give gel (ionique gelation and film hydration method) or liquid products (complexation, emulsification, and coacervation) [1]. Another important point is the practical parameters of the techniques—spray drying needs specific and expensive equipment, but is very easy and fast [51], while ionic gelation needs simple products but with more manipulations [78]. The choice of a technique has to be done depending on the final product application.

Another observation of this review is that some articles mention that a lot of factors influence the efficiency and the release profile of encapsulations. In order to develop the use of EOs at an industrial scale, these factors have to be optimized for the encapsulation of each product of interest, including the analysis of the physical interactions and behavior through specific analytical means.

Considering that an EO is a complex mixture of compounds, each with a different volatility, it seems important to deepen the study of the released chemical profile in order to control the activity as well. Some compounds have a strong bioactivity, others do not, and synergy can also be frequently observed [94]. Thus, to develop an efficient biosourced pesticide, we have to control whether the profile of the released compounds is constant over time.

The last proposition for a suitable future study about EO encapsulation for controlled release is to find new stable, biosourced, and economically viable matrices that could entrap EOs.

Finally, it is important to note that before any use in the field, ecotoxicity studies should be carried out to assess the impact of essential oils, but also of encapsulation matrices on non-target organisms. Particular attention has to be given to decomposition products and persistence in the environment in order to avoid the accumulation of unwanted products.

In conclusion, considering the high number of studies about EO encapsulation, the interest in extending the activity life of EOs is obvious, but it is time to go further with methods and products that have an applicative interest. There is a real challenge to create competitive products that also respect the environment.

## Figures and Tables

**Figure 1 molecules-24-02539-f001:**
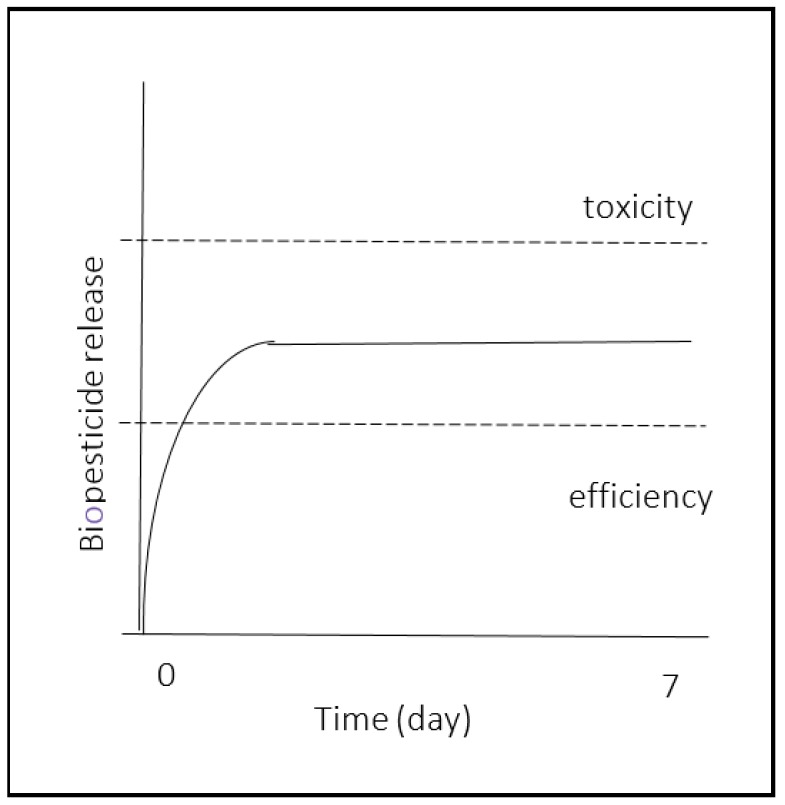
Controlled release of biopesticide outline showing an initial rapid release at the middle concentration between the active agent’s efficiency and toxicity scale, followed by a long and constant release.

**Figure 2 molecules-24-02539-f002:**
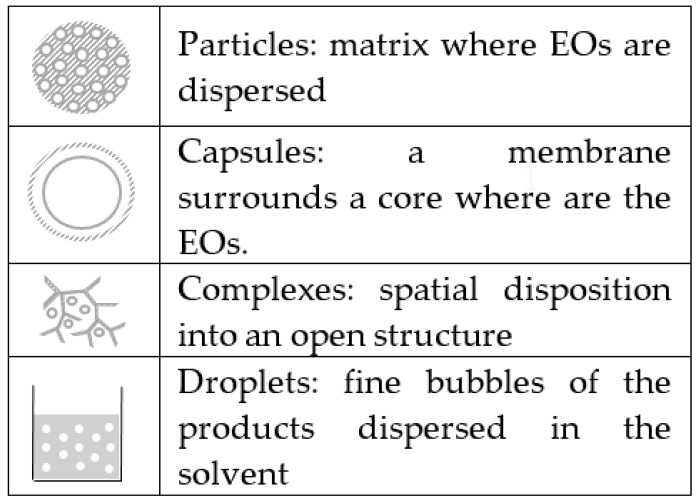
Illustration of essential oil (EO) encapsulation types.

**Table 1 molecules-24-02539-t001:** Encapsulation techniques of essential oils (EOs) allowing a controlled release.

Encapsulation Method	Capsule Type	Encapsulation Preparation	Capsule Size	Examples of Matrix	Ref.
Emulsification	Droplets	Simple emulsion: Spontaneous emulsification method*. The organic phase is added to an aqueous phase, then stirring at an ambient temperature (∼25 °C) for 15 min. →Nanoemulsion	75 nm	Organic phase: oil carrier (medium chain triglyceride), EO, and surfactant (tween) Aqueous phase: citrate buffer	[22,43]
		High Pressure homogenisation (HPH) technique: Primary emulsions obtained by stirring at 24,000 rpm for 5 min are then subjected to HPH ten times at 350 MPa. Final step: crystallisation by rapid cooling in an ice bath. →Nanoemulsion	75–175 nm	Organic phase: oil carrier (sunflower oil or palm oil), EO, and surfactant (soy lecithin, tween, or glycerol mono-oleate) Aqueous phase: water	[44]
	Capsules	Multiple emulsion: Emulsion of aqueous phase 1 in organic for 10 min at 800 rpm, then second emulsion in aqueous phase 2 at 500 rpm (w_1_/o/w_2_). Stirring is continued for 3 hours to allow for solvent evaporation. Finally, microcapsules are filtered, washed, and air-dried overnight at room temperature. →Microemulsion	200–400 μm	Carbohydrate polymer–protein blends: Aqueous phase 1: sodium alginate Organic phase: EO, methylene chloride solution of ethyl cellulose, and surfactant (tween) Aqueous phase 2: gelatin and surfactant (tween)	[45]
		Interfacial mini-emulsion: Sample emulsion (organic and aqueous phase stirring for 10 min) followed by ultrasonication for 3 min under ice cooling. End step consists of a gelification process by 20 hours stirring.	100 nm	Tetraethyl orthosilicate	[46]
Coacervation	Capsules	Simple methodology: Addition of a solvent to a hydrophilic colloidal solution at 20 °C with stirring. →Coacervates	>1 μm	Solvent: sodium sulphate solution, acetone, or alcohol Hydrophilic colloidal solution: gelatin solution	[1]
		Complex methodology*: Dispersion of EO in gelatin at 50 °C for 3 min at 14,000 rpm followed by the addition of Arabic gum at the same temperature. Then the mixture is cooled at 10 °C for 16 hours under stirring. Finally, reticulation by TPP is performed at room temperature for 2 hours.	40 μm	Wall = gelatin and Arabic gum solution Core = EO and sodium tri-polyphosphate (TPP)	[47] *[48]
Spray drying	Particles	Liquid atomization intro small droplets: Emulsion oil/water (O/W). Dissolve matrices in water at 50 °C for 2 hours. Add oil phase by stirring at 24,000 rpm for 30 min. Spray dried. Emulsions are atomized by a hot air stream in the drying chamber of a spray dryer.	0.2–40 μm	Arabic gum and maltodextrin	[49,50]
			3–4.5 μm	Inulin solution to make Raftalin microparticles	[51]
			223–399 nm	Alginate and cashew gum	[52]
			9–15 μm	Chitosan, chitosan and alginate, and chitosan and inulin	[53]
			12–13 μm	Modified starch and Arabic gum	[54,55]
			28–435 μm	Cashew gum	[56]
Complexation	Complex	Spontaneous complexation reaction: Add EOs to a cyclodextrin aqueous solution and thermostate at 25 °C for 30 min.	/	β-Cyclodextrin and derivatives	[57,58,59,60,61,62,63]
		Co-precipitation: Add EOs to a β-cyclodextrin solution at 55 °C under stirring for 4 hours, then cool at 4 °C overnight. Complexes are obtained by filtration and drying at 50 °C for 24 hours.	/	β-Cyclodextrin	[64,65,66]
		Freeze-drying: Add EOs in alcohol to an aqueous solution of cyclodextrin, stirring at 180 rpm for 7 days at 37 °C, and freeze-drying the filtered solution to obtain solid complexes.	/	β-Cyclodextrin	[67,68,69,70,71,72,73]
Ionic gelation	Particles	Two step method: Emulsion O/W. EOs are added to the sodium alginate solution, and the mixture is stirred at 300 rpm. Ionic gelation by crosslinking with divalent ions. Calcium chloride is added to the emulsion stirring at 300 rpm at 30 °C for 30 min.	20 μm–1 nm	Alginate Crosslinker: Calcium chloride	[74]
		Emulsion O/W. EO are added in an aqueous chitosan solution at room temperature, and stirred vigorously for 30 min. Ionic gelation by adding a crosslinker in the emulsion and stirring for 1 hour at room temperature.	235 nm	Chitosan crosslinker: pantasodium tripolyphosphate (TPP) and sodium hexametaphosphate (HMP)	[23]
			30–80 nm		[75]
			125–175 nm		[76]
			140–237 nm		[77]
		Three step method: Single emulsion O/W. Add EO in an alginate aqueous solution at room temperature with stirring at 13,500 rpm for 5 min. Multiple emulsion O/W/O. Add a primary emulsion in the oil phase under stirring at 10,000 rpm for 3 min. Ionic gelation reaction. Add calcium chloride dropwise to an O/W/O emulsion under stirring for 20 min.	47–117 μm	Alginate Crosslinker: calcium chloride	[78]
Nanoprecipitation	Nanoparticles	Dissolution of polymers and EO in acetone, followed by stirring in aqueous phase with a surfactant for 10 min.	210 nm	Poly(DL-lactide-co-glycolide) (PLGA)	[79]
	Nanoparticles	Addition of an acid solution of chitosan to a methanol EO solution under moderate stirring at room temperature.	3 μm	Chitosan	[24]
Film hydration method	Nano-cochleates	Liposome prepared with phospholipids, cholesterol, and EO by stirring in an organic solvent are then dried and rehydrated by phosphate buffered saline. This dispersion is then stirred in water bath at 37 °C for 30 min. Trapping method with divalent cation. The calcium chloride solution is added dropwise to a liposomal suspension under stirring at 150 rpm at an ambient temperature for 15 min.	250 nm	Divalent cation: Calcium chloride	[80,81]
	Multilamellar Vesicles	Spontaneous formation of vesicles by the hydration of an organic phase containing EO and phospholipids for 2 hours in the dark at room temperature.	0.5–100 nm	Based on phosphatidylcholine, cholesterol, and calcium ions	[1,82]
Other	Active film	Emulsion. Add EO and sorbitol to the alginate solution under strong stirring for 15 min. Ionic gelation. Add calcium carbonate to the emulsion and adjust pH to 4.0. Add of a plasticizer. Casting the emulsion at 40 °C for 15 hours.	/	Alginate Crosslinker: calcium carbonate Plasticiser: sorbitol	[83,84,85]
	Nanogel particles	Formation of amide linkages through an EDC-mediated reaction* (1-Ethyl-3-(3-Eimethylaminopropyl) Carbodiimide). Chitosan aqueous solution is added to the EDC and caffeic acid solution and stirred for 5 hours. Nanogel is precipited by adjusting the pH at 8.5–9 using sodium hydroxide.	≤100 nm	Chitosan–caffeic acid nanogel	[86]
				Chitosan–cinnamic acid nanogel	[87,88]
	Particles	Rapid expansion of supercritical solutions (RESS) in a reactor. Mixing of the EO, liposomal material, organic, and supercritical solvent under pressure for 1 hour, followed by the addition of phosphate buffered saline. Fast spraying of the mixture into the collector to evaporate supercritical fluids.	173 nm	Liposomal material Supercritical carbon dioxide fluid (scCO_2_)	[89]
		Rapid ultrasonication method: Primary chain of starch and EO are dispersed in water and irradiated by the ultrasonic horn at room temperature for 10 min to form nanoparticules.	200 nm	Starch	[90]
		Plasmolyze yeast cell and then fill them by diffusion through the cell membrane pores (three times a day at 40 °C).	9 μm	Baker’s yeast	[91]

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
