# Peer review of "Encapsulation of Essential Oils for the Development of Biosourced Pesticides with Controlled Release: A Review"

_molecules, 2019, doi:10.3390/molecules24142539_

Round 1

Reviewer 1 Report

In the paper title “Encapsulation of essential oils for the development of  biosourced pesticides with controlled release: a  review” authors provide a good review on encapsulation methods for essential oils. There are some points that needs further explanations.

1)    In line 38 of introduction, authors say that EOs obtained by ScCo2 or microwave assisted extraction, cannot be considered as Eos even through this processes are widely used this years. Could the author clarify how these oils should be named?

2)    In line 42 “the largest component” should be changed to “The main component”

3)    Lines 70 and 71, should be rewritten and give further explanations on how this encapsulations techniques accomplished the control of Eos high volatility, since this information is not found in the rest of the paragraph.

4)    In line 102, there is a reference to three types of encapsulation but in figure 2 there are four types listed, which data is correct?

5)    Table 1 should provide more data, such as processing times, temperatures… in order to allow the readers to carry out these processes in his lab.

6)    The paragraph between lines 138 and 142, should provide information about how the “in situ” and “in vitro” assay are carried out

7)    Alginate is found more than three times in table 1 author should correct line 158

8)    In line 173 reference 79 is not included in table 1. Author however say that this reference is one of the first methods where chitosan Is used as a matrix

9)    In lines 179, 180 authors talk about a method without adding any reference, I think this reference is important in order to give full information on this method

10) In point 3.3 references used in the text do not match the references used in table 1

11) In point 4, discussion, authors talk about matrices but do not talk so much about techniques. Can the authors give more information about the techniques?

Reviewer 2 Report

Well done, indeed. I really liked Your valuable manuscript (MS). Accept after minor revision.

English language and style are fine/minor spell check required.

In addition to this, You may kindly consider citing of the following references within Your MS:

Cryptogamie, Bryologie Volume 32, Issue 2, April 2011, Pages 113-117

- Industrial Crops and Products, Volume 49, August 2013, Pages 561-567

Last but not least, very best of (research) luck ahead!

Reviewer 3 Report

The article presents a survey of techniques for encapsulation of plant essential oils, for their use in different areas. I find the article well written and presented. I note a few minor points that should be improved in the manuscript before publication.
Figure 1 is not well described and clarified in the text nor in its legend, so both skilled and not-skilled readers could find it useless, although for opposite reasons.
The Authors refer applications of encapsulated EOs, as in cosmetics, and for herbicides and pesticides. Differences among the products of the encapsulated techniques should be considered on the basis of the application. Is any encapsulation considered safe in cosmetics, not causing allergy? Similarly, the environmental impact should be discussed, in particular in the usage of EOs as herbicides and pesticides in open field conditions. The use of natural polymers does not exclude the risk of eccessive immission in the nature of unexpected materials, as well as the pollution by nanoparticles is a typical problem related not only to the nature of the material. Comments on these aspects should be added under Discussion section or other appropriate paragraph.
